# Sustainability in Heritage Buildings: Can We Improve the Sustainable Development of Existing Buildings under Approved Document L?

Andrew Williamson [1,*] and Stephen Finnegan [2]

1   Pollard Thomas Edwards Architects, London N1 8JX, UK
2   Zero Carbon Research Institute (ZCRI), School of Architecture, University of Liverpool (UoL), Liverpool L69 7ZN, UK; s.finnegan@liverpool.ac.uk
*   Correspondence: andrew.williamson@ptea.co.uk; Tel.: +44-77-2289-2406

**Abstract:** The British government has ambitions to achieve net-zero carbon emissions countrywide by 2050, with their largest challenge being emissions from the construction industry. Approved Document L sets standards for the fabric performance of buildings to regulate their consumption of fuel and power, thereby allowing easier transition to all-renewable grid electricity and the phasing out of fossil fuels. Whilst this approach has shown success in new builds, its effectiveness on retrofits, especially regarding built heritage, is significantly reduced. Responding to this, the paper investigates alternative sustainable design solutions that could feasibly justify revisions to Approved Document L, to improve the sustainable performance of existing buildings and bring them in line the government's 2050 targets. Trialing solutions on a listed building case study, benchmark figures are used to analyse the energy, carbon, and cost implications of sustainable design approaches relating to passive design, low-carbon technologies, renewable energy, and additional considerations. Using this method, it is reasonable to conclude that the standards of Approved Document L for existing buildings are currently underperforming but can feasibly be revised to encompass the full breadth of contemporary sustainable design solutions.

**Keywords:** heritage; architecture; retrofit; listed; building; construction; sustainable design; Approved Document L; BREEAM



## 1. Introduction

Climate change poses one of the greatest influences on contemporary building design. A 2018 report by the UN declared that we must extinguish global greenhouse gas emissions by 2030 or suffer an average 1.5 °C temperature rise, and the irreversible climate impacts that will follow [1]. Responding to this, the British government has announced their ambitions to achieve net-zero emissions by 2050 [2], meaning all greenhouse gas emissions must be eliminated or negated through positive offsets.

In the UK, $CO_2$ accounts for 81% of total greenhouse gas emissions and the construction industry, including buildings in operation, accounts for approximately 45–50% of all carbon emissions [3,4]. Approved Document L (hereafter known as 'Part L') addresses sustainable development through the conservation of fuel and power in operating buildings. This is achieved by enforcing fabric performance standards on new and existing building designs and regulating the efficiencies of mechanical systems, such as heating and ventilation. Ultimately, this aims to reduce energy use in buildings enough to allow transition to entirely renewable grid electricity whilst phasing out fossil fuels.

Despite these ambitions, there is still a fundamental divide in the standards for new and existing building projects under Part L. For wall performance in non-dwellings the threshold thermal transmittance in existing structures is half that of new builds at $0.7 \, \text{W/m}^2$ and $0.35 \, \text{W/m}^2$, respectively [5,6], essentially permitting them to lose twice as much energy

through heating. This gap is further exacerbated by the new 'Future Homes' legislation which looks to significantly improve the low-carbon performance of new dwellings whilst neglecting existing dwellings and non-dwellings [7]. Only ~11.3% of homes in England have been built since 2000 [8,9]; assuming a similar value is true across the entire UK, we can clearly see that these updates will not impact most homes by 2050 and will have no direct benefit for non-dwellings.

Additionally, English Heritage—a non-departmental public body that denotes the listing of heritage structures—allow the standards of Part L to be jeopardised if there is an argument in favour of protecting the historic significance of existing fabric when retrofitting listed buildings [10]. Though this has the ambition of protecting British architectural heritage, the decision-making process is largely subjective, leading to a bias against the implementation of sustainable development measures that render the Part L standards impotent and again highlighting the problems a fabric-first design approach can pose for existing buildings [11].

Another issue with the current state of these standards is their lack of consideration for the lifespan of buildings. In their 2010 estimation of the construction industry's carbon footprint, HM Government acknowledged that 17% of emissions occur during the design, manufacturing, construction, and demolition stages of building projects [12]. As such, addressing the energy consumption of buildings in operation alone neglects a large portion of lifetime emissions and presents a misguided attempt at achieving zero-carbon. Moreover, the figure of 17% was calculated in 2010 but, as operating efficiencies progress, we can see embodied carbon grow as large as 70% of overall emissions in contemporary designs [13]. Therefore, Part L currently fails to fully address the carbon footprint of buildings.

Outside of the Building Regulations, BREEAM is one of the UK's most popular sustainability rating tools, offering universal marking criteria under an expanded set of considerations. The major benefit of this is that the criteria are holistic, considering all lifetime emissions. A report published by BREEAM has further shown that, despite having similar assessment criteria, retrofits and listed buildings are able to perform as well as, or better than, new builds in several of their assessment categories (Figure 1) [14], not least for their reduced consumption [15]. In many cases these retrofits are also able to achieve Excellent or Outstanding accreditation, BREEAM's highest awards. Whilst the justification for splitting the standard between new and existing buildings in Part L can only be assumed, the fact that BREEAM does not draw a similar distinction, whilst maintaining high performance in completed projects, accentuates the misguidance of Part L's approach.

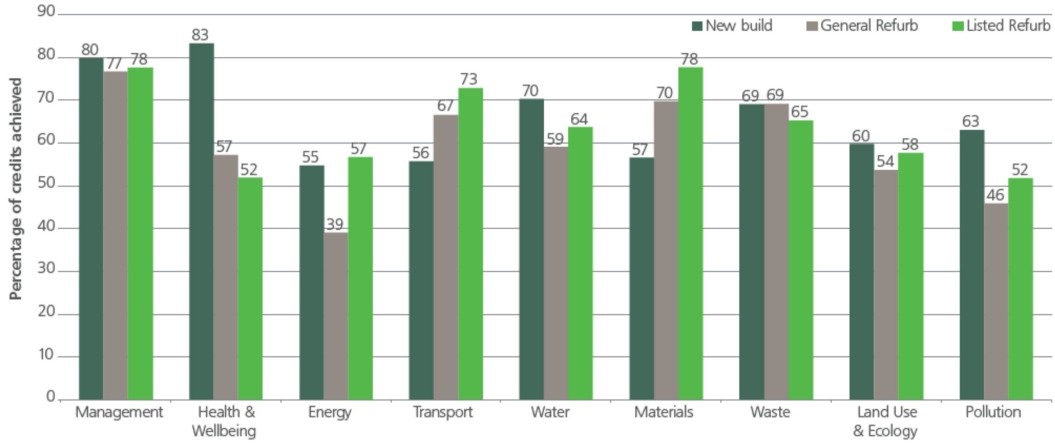

**Figure 1.** Credits attained by new builds and retrofits, by category, under BREEAM; © by BREEAM.

Overall, there is strong evidence to suggest that the Part L standards for existing buildings are underperforming, which brings into question the suitability of its current approach to achieving zero carbon. Greenwood et al. and Salem et al. supported this

claim in their review of low and zero carbon homes in England and the retrofit of UK residential property to achieve zero-energy buildings [16,17]. The policy implication for the gap in building technology vs. the Part L calculations has also been considered by McElroy and Rosenow [18]. Moreover, Shen et al. have stated how policy instruments for improving building energy efficiency need to advance [19], with Tombs discussing the positive impact good regulation can have towards improving building performance [20], showing that an improvement to the existing building standards of Part L could reap significant environmental benefits across the country.

To this end, the study analyses the effectiveness of current standards for existing buildings under Part L and proposes alternative sustainable development strategies that show promise in reducing their environmental impact. Yung et al. have previously speculated the relevance of economics, cultural heritage and environmental preservation as factors in the sustainable development of historic buildings and, respecting those themes, this study expands upon the energy and carbon focus of Part L to consider design impacts and financial feasibility of proposed solutions [21]. This has the wider ambition of identifying effective sustainable development solutions for existing buildings to inform a framework of revisions that could improve Approved Documents L1b and L2b, reflecting the full breadth of sustainable design technologies available in the modern construction market. In this way we can ensure that British buildings, existing or new, are doing all that they reasonably can to achieve higher standards of sustainable performance, therefore reducing the environmental impact of the British construction industry.

As such, the study will address three main objectives: (i) To identify the effectiveness of Part L's current fabric-first design approach in reducing the energy demand of existing buildings; (ii) To identify financially viable alternative methods that could reduce the energy and carbon consumption of existing buildings, other than Part L's fabric-first approach; (iii) To explore the potential of additional low-carbon design solutions in achieving retrofits with low environmental impact, as exhibited by alternative assessment tools such as BREEAM.

## 2. Materials and Methods

* A list of technical definitions and acronyms can be found in Appendix A.

### 2.1. Defining Sustainability

To quantify sustainable design, it must first be defined. Traditionally, environmental sustainability is "*the degree to which a process or enterprise is able to be maintained or continued while avoiding the long-term depletion of natural resources*" [22]. However, HM Government's official understanding stems from the Intergovernmental Panel on Climate Change (IPCC) definition presented in Agenda 21, which states that sustainable development is "*making the necessary decisions now to realise our vision of stimulating economic growth and tackling the deficit, maximising wellbeing and protecting our environment, without negatively impacting on the ability of future generations to do the same*" [23]. Although this definition also relates to a social dimension, in the context of this paper we will focus on the environmental and financial implications of sustainable design solutions when analysing their effectiveness.

### 2.2. Methodology

To rationalise the impact that various sustainable design interventions can have on retrofits, the research will consider two case studies of retrofitted listed buildings. They have been chosen for two key reasons: because of the heightened issues that can afflict heritage assets versus standard retrofits, acting as extreme examples of the general considerations for sustainable design in existing buildings; and for the contrasting levels of sustainable performance attained by the two projects.

The Edinburgh Centre for Carbon Innovation (ECCI) became the first listed building in the UK to achieve BREEAM *Outstanding* accreditation after its completion in 2013, whereas 30 James Street Hotel struggled to comply with the Part L benchmarks and continues to

suffer from high energy consumption. The two buildings both achieved grade 2* listing for their respective histories, were completed 6 months apart, and are non-dwellings, therefore being comparable under the same baseline Part L performance standards. Moreover, both buildings are located in British urban centers within UNESCO world heritage sites. It should be acknowledged that the ECCI received a higher budget for the scope of its works, although it is not the agenda of this study to speculate what can be achieved with larger capital investment, but rather propose effective sustainable design strategies that also provide economic value.

Analysing the ECCI as a best-case example of sustainable design in the retrofit of an existing building enables effective sustainable design strategies to be identified. Speculating the effectiveness of implementing similar strategies at 30 James Street allows for potential energy and carbon savings to be quantified and observed in comparison to the existing demand of the building in operation (Figure 2). Although Part L only considers the energy efficiency of building systems in its benchmarks at present, understanding carbon savings allows for a better comparison between solutions that are energy dependent and otherwise, whilst also helping progress the unit of sustainable design from energy demand into sequestered carbon. Furthermore, as per HM Government's sustainable development definition, capital costs, savings, and investment return periods will be determined, culminating in a set of effective carbon saving and economically viable sustainable design proposals for 30 James Street.

In rationalising that these solutions could have been feasibly implemented at 30 James Street, without adverse effects on project financing, evidence is provided that the existing buildings standards of Part L are underperforming and should be revised. Moreover, the solutions identified will inform the basis of the framework of revisions that would more appropriately align the standards with HM Government's zero-carbon targets.

Data Collection

Information regarding the case studies has been collected through a variety of sources, including online research, first-hand site visits, and interviews with building management. Additionally, formal liaisons with the respective designers of each case study (Signature Living for 30 James Street; Calum Fraser Architects for the ECCI) led to the procurement of planning reports and supporting documentation, upon which the case study analysis is primarily based.

The carbon and cost implications of proposals have been calculated predominantly with figures found in the report "Improving Historic Soho's Environmental Performance: Practical Retrofitting Guidance" published by the City of Westminster. This document utilises data collected from a series of refurbished listed buildings in Soho, London to identify trends in the performance of various sustainable design solutions and rationalise them into energy, carbon, and cost benchmarks. This data is further supplemented, where necessary, with relevant product data taken from suppliers and manufacturers, including concurrent gas and electric supply rates. Furthermore, estimates of the existing energy and carbon demands of 30 James Street have been calculated using the CIBSE TM46 Benchmarks, providing baseline figures against which projected savings have been analysed.

## 2.3. Analysis of the Case Studies

30 James Street (Figure 3) was originally built as headquarters for the White Star Line. After their ship RMS Titanic sank, commemorative plaques for the deceased were pinned to the façade of the building, giving it its listed status. The building was severely bombed during the Second World War, but remained structurally sound, with much of the façade intact. The modern retrofit was the brainchild of local Lawrence Kenwright and was only made possible through Singaporean investment and a tight schedule. Combined with stringent limitations on the entire external aesthetic, this led to difficulty achieving the thermal performance benchmarks of Part L. As a result, the building still suffers from high energy consumption and costly bills.

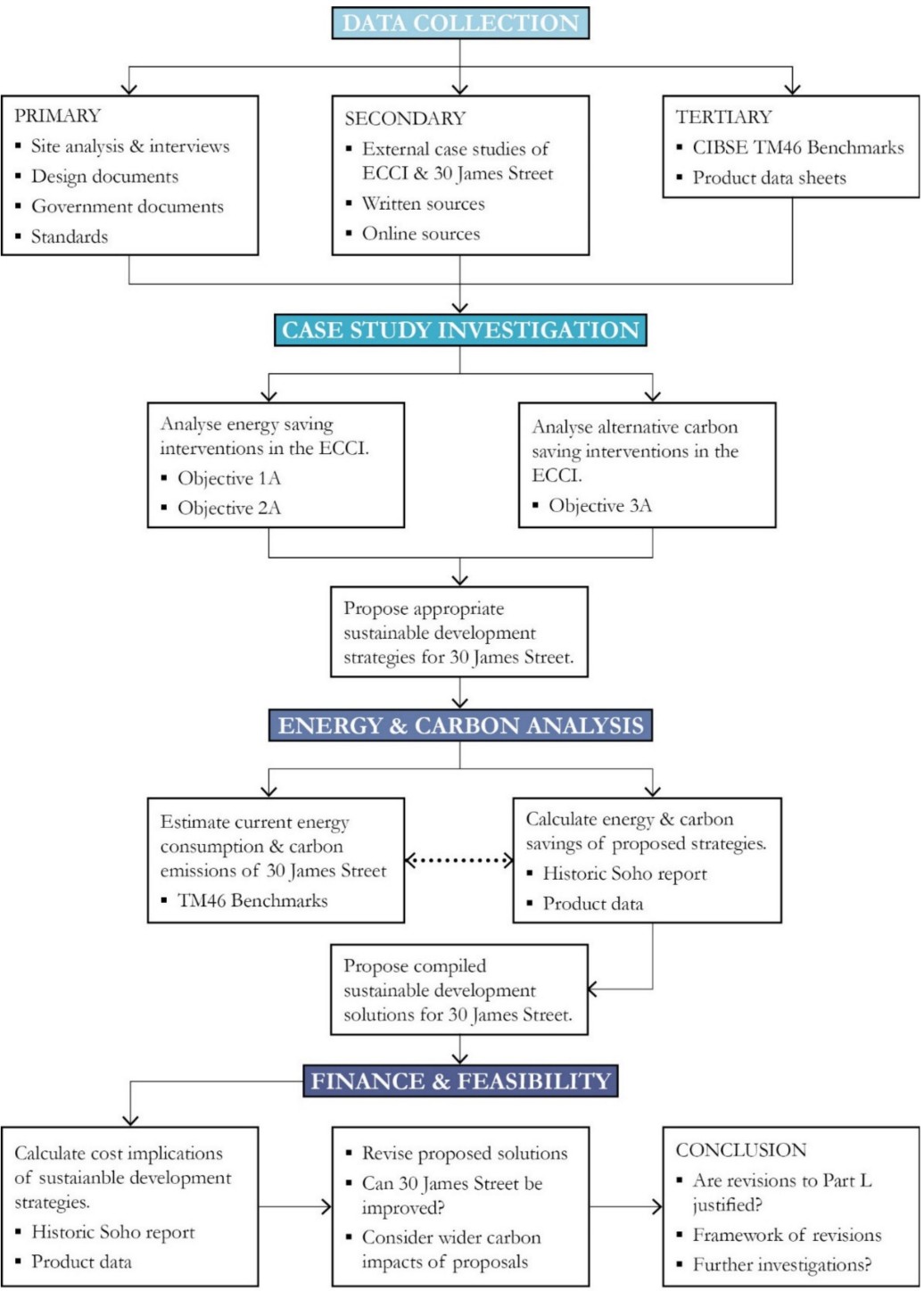

**Figure 2.** Diagram showing methodological approach to research and investigation; author's work.

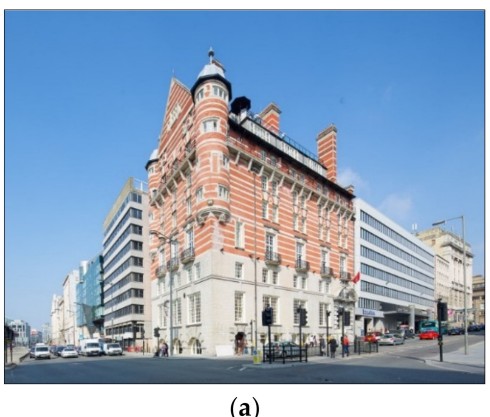 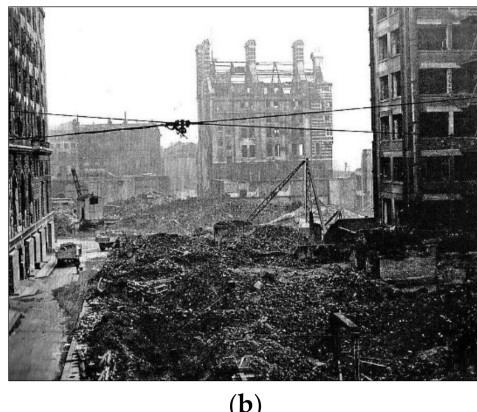

(**a**) (**b**)

**Figure 3.** 30 James Street in its context: (**a**) Contemporary building after refurbishment, 2019; (**b**) 1940s resultant structure remaining after Liverpool air raids; © by Signature Living.

Contrastingly, the ECCI (Figure 4) achieved BREEAM *Outstanding* accreditation for both design and construction stages. Sustainability was a primary concern from inception, and the modern design, which knits together two pre-existing structures, functions as a low-carbon innovation hub. The site has history stretching back to the 12th century when it was founded as a monastery. It has changed hands and functions over the years, but the modern structures evolved from a 16th century school and 19th century surgical hospital. Despite this, only the front elevation of the main building on High School Yards is protected, allowing much of the exterior to be adapted and connected to a new extension.

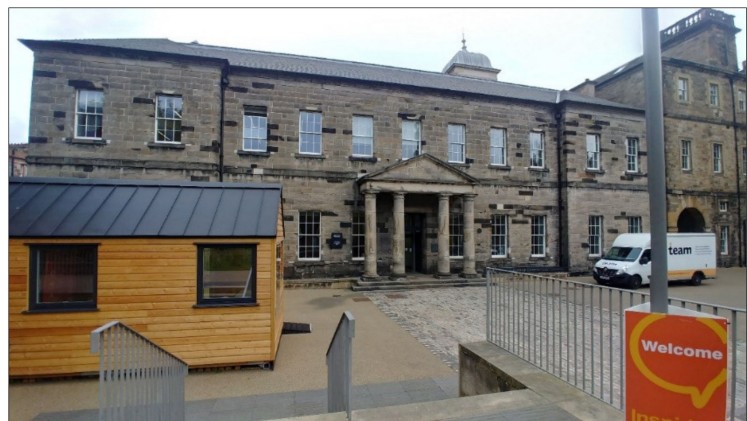

**Figure 4.** Protected main elevation of the Edinburgh Centre for Carbon Innovation on High School Yards; author's photograph.

### 2.3.1. Passive Design

The ECCI operates on a very low energy demand due to a mixture of high thermal performance in the building envelope and integrated low-tech passive systems. As only the building exterior was protected, the internal envelope was pliable to significant thermal improvement. Resultantly, the build-up lowered wall heat losses to 0.25 W/m$^2$ K, far below Part L's benchmark of 0.7 W/m$^2$ K. In the overheating months, controlled opening vents in the central atrium provide a low-tech method of removing excess heat, whilst smaller rooms make use of large sash windows for manual ventilation (Figure 5). Conversely, the large, retained windows from the original structures, in addition to extensive new glazing, mitigates a portion of winter heating demand by encouraging natural solar gains.

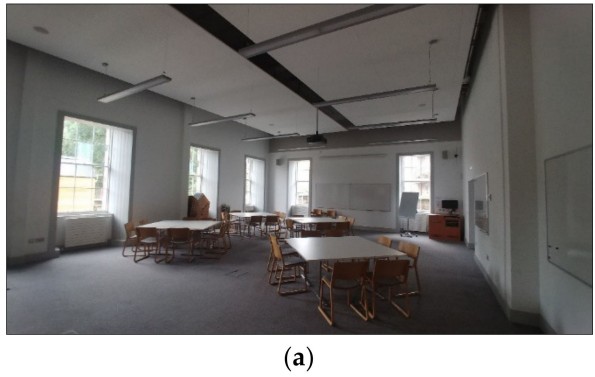
(**a**)

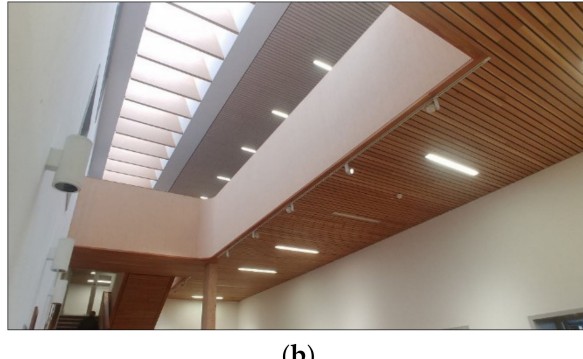
(**b**)

**Figure 5.** Passive design in the ECCI (**a**) Retained large sash windows from original structures providing daylighting and solar gains; (**b**) Diffused daylighting and electronically controlled opening vents in the ECCI central atrium; author's photographs.

At 30 James Street the biggest challenge encountered during the retrofit was updating thermal performance without damaging unique original features, such as the mosaics and riveted steel structure. Many attempts at retaining this character resulted in compromised sustainable design solutions, or lack thereof, most notably in the packing of insulation beneath external turrets, displaying structural timbers but at the cost of creating significant thermal bridges (Figure 6). Similarly, a desire to preserve the existing entrance foyer led to inefficient space heaters becoming the primary heating source.

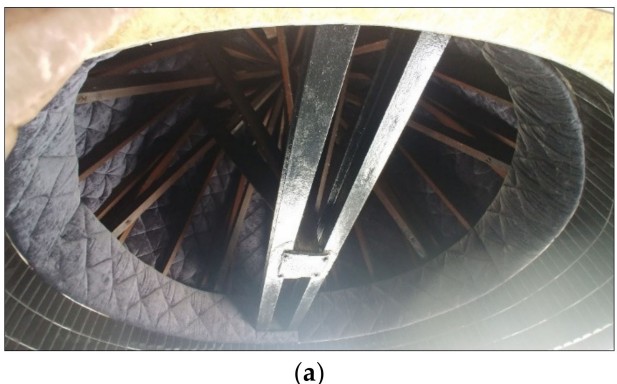
(**a**)

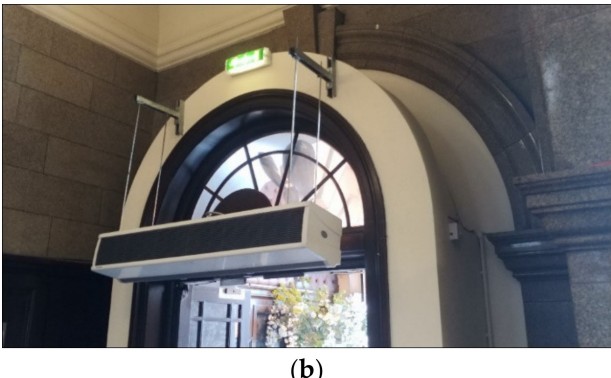
(**b**)

**Figure 6.** Thermal performance at 30 James Street (**a**) Insulation packed between existing timber rafters beneath a turret; (**b**) Space heaters providing primary heating in the entrance foyer; author's photographs.

Despite these limitations, there are some design solutions from the ECCI that might be implemented at 30 James Street. Existing doors and windows could be updated with high performance modern equivalents and draught proofing to improve thermal retention. The ECCI was able to retrofit sash panes with slimline double glazing, which has been attempted in the basement spa at 30 James Street, but primarily for condensation management. Alternatively, reinstating window shutters has also been shown to improve heat retention to a similar degree with reduced upfront costs [24].

### 2.3.2. Low-Carbon Technologies

Aside from envelope performance, the ECCI utilises LEDs throughout reducing lighting energy consumption to 10% compared with the traditional fluorescent and incandescent bulbs that are often found in older buildings [25]. As LEDs tend to be brighter and more durable, they also require reduced maintenance and fewer replacements over time. Aside from lighting, the ECCI employs more advanced technologies to reduce mechanical heating and cooling demand in the form of chilled beam cooling and Air Source Heat Pumps

(ASHPs). Essentially these systems transfer ambient heat from external air into heating and cooling elements in the larger internal spaces. Heat exchangers are also employed, recycling energy between inflowing and outflowing air currents to maintain indoor thermal equilibrium, improve the operating efficiencies of the systems and reduce heating energy demand.

Whilst 30 James Street is also supported by structural steelwork, it is largely encased and could not be used for chilled beam cooling. ASHPs, however, show potential to provide a renewable heating strategy throughout the building, replacing the need for space heating; combining them with heat exchangers could see significant energy reductions for heating. Additionally, LED lighting can be easily implemented to drop lighting energy demand with little upfront cost.

### 2.3.3. Renewable Energies

A key strategy in reducing energy demand, currently neglected by Part L, is on-site renewable energy generation, which the ECCI employs in the form of a Combined Heat & Power (CHP) system. The system is not housed directly on-site but is connected to the University of Edinburgh's district heating network, providing 56% of the ECCI's energy demand and depreciating carbon emissions by 38% [26]. CHP units are especially efficient as waste heat from electricity generation is recovered to supply hot water and ambient heating systems. The remaining 44% of the ECCI's energy demand is supplemented with 30 m$^2$ of solar Photovoltaics (PVs), the effectiveness of which is maximised by strategic south-facing placement.

30 James Street presents a more complicated case for renewable energy, as the external building aesthetic is strictly protected in Liverpool's Maritime conservation area, although there is scope to replace existing heating infrastructure in the basement with a micro-CHP boiler. The University of Edinburgh powers their CHPs with fuel cells, a new and expensive renewable source that chemically converts hydrogen and oxygen in electricity and water. However, alternative fuels, such as biomass, could prove cheaper and more suitable. There is also potential for a glazed roof terrace to be fitted with solar PVs. Although tradition panels would not be accepted, Polysolar (Cambridge, UK) offers a range of transparent photovoltaic glasses that could be installed along ~116 m$^2$ of exposed glazing with a southern aspect (Figure 7) [27].

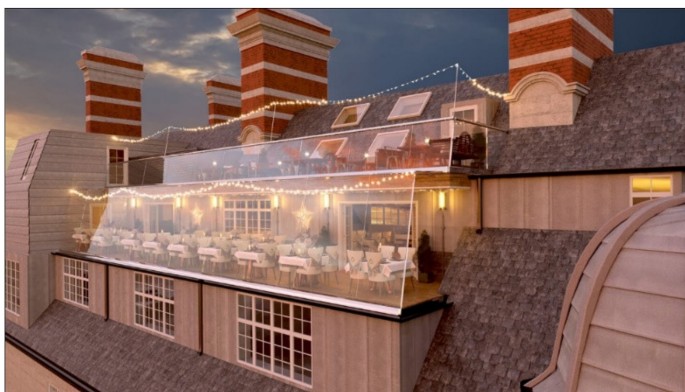

**Figure 7.** CGI showing the proposed glass roof terrace at 30 James Street; © by Signature Living.

Alternatively to on-site renewables, many power suppliers offer opt-in services for purely renewable energy supplies. In these contracts power still comes from the grid, but energy bills fund the development of new renewable energy generation. SSE supply one such scheme authenticated by Ofgem (HM Government's gas and electricity regulator) [28]. Additionally, companies such as The Poseidon Foundation use carbon offsetting to counterbalance the emissions of a project by funding carbon-positive agendas worldwide, including reforestation and the supporting of local communities to reduce dependence on

deforestation jobs [29]. Although they are valid approaches to counteracting the carbon emissions of buildings, off-site renewables and carbon offsetting should only be resorted to once other pathways have been explored, as they do not tackle the fundamental issue of sustainable development—to reduce carbon emissions.

2.3.4. Additional Considerations

In addition to energy management in the forms of passive design, low-carbon technology, and renewables, BREEAM offers an expanded set of considerations looking at the indirect emissions of construction. This is addressed under the categories of materiality, pollution, transport, waste, water, health and wellbeing, and management. There is also the category of Innovation, although this is assessed on a case-by-case basis and does not factor into consideration here.

Materiality

As discussed in the introduction, embodied carbon can account for as much as 70% of whole life carbon in modern buildings, which can be predominantly addressed in the materiality and construction methods of a building project. The greatest asset of existing buildings over new builds is their ability to reuse built fabric, creating minimal embodied carbon through the manufacturing of new materials. Much of the ECCI's existing fabric was retained, including the structure and cladding of external walls and floors. Construction of the new extension was further built on a timber frame, and much of the interiors were clad in timber from renewable northern European forests (Figure 8). The major benefit of this is that wooden products are often carbon-positive, absorbing more $CO_2$ in their growth than released during their manufacture, installation, and dismantling. The Cross Laminated Timber (CLT) structure was estimated to have captured 4–5 times more carbon than emitted [26]. Furthermore, the local sourcing of timber and stone considerably reduced transport emissions, contributing to a lower embodied carbon. This attention was further extended to the fittings, incorporating all steel power outlets and timber furniture, amongst other efforts.

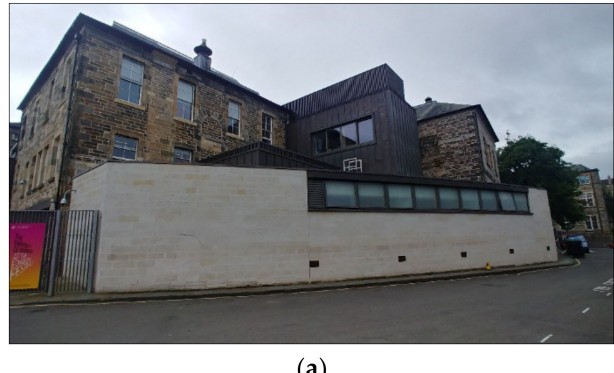 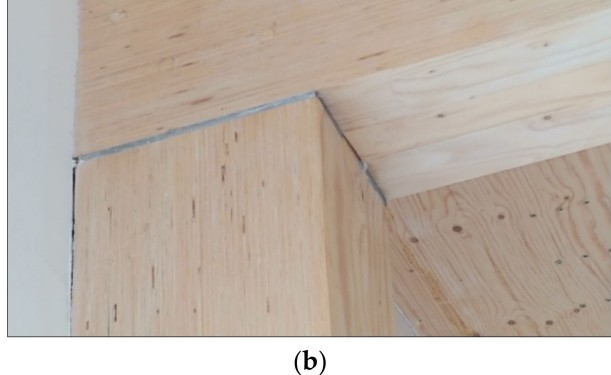

(**a**)  (**b**)

**Figure 8.** ECCI materiality; (**a**) Locally sourced Fife stone in the new extension, reflecting the retained materiality of the existing structures; (**b**) Timber and steel structural joint in the new extension; author's photographs.

Materiality is one of the categories in which 30 James Street can be thought of as more sustainable. Though the project was a retrofit, much of the existing building was retained and celebrated (Figure 9). Although this did have ramifications on insulation, the relatively light touch of the project required little mass of new materials and finishes, culminating in little embodied carbon.

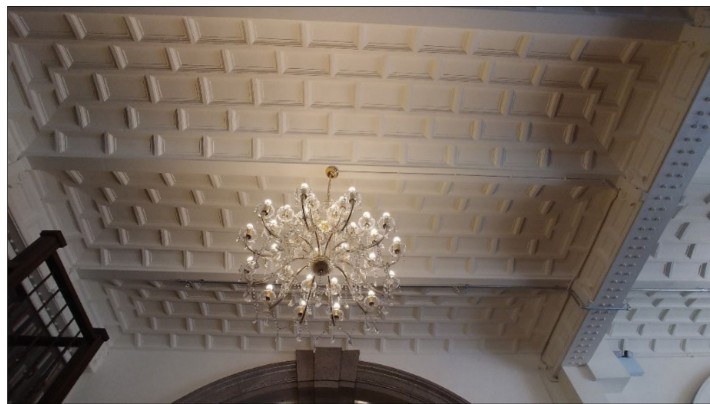

**Figure 9.** Original ground floor ceiling and steel structure retained and exposed at 30 James Street.

Management, Health & Wellbeing

Often, we find there is a disconnect between designed performance and actual performance in sustainable buildings, mostly due to mismanagement of low-carbon systems. The ECCI uses meters and displays to monitor energy-use throughout the building. This was designed to be educational, but also serves as a useful tool in identifying excessive energy from appliances and heating, to be streamlined by management staff. A report by Sturgis Carbon Profiling highlighted that this kind monitoring system can facilitate as much as a 5% reduction in energy consumption because of improved building management [24].

Land Use & Water

Environmental sustainability regards the protection of all-natural resources, not solely carbon emissions. Flora plays an important role in this regard by encouraging biodiversity in urban environments whilst providing a means of carbon-capture. The ECCI planted trees and wild grasses in the grounds and on green roofs to this end (Figure 10), with the added benefit of providing sustainable urban drainage and evaporative cooling to combat the urban heat island effect. As a result, the ECCI's cooling demand is reduced in summer and the wellbeing of people in the grounds is improved.

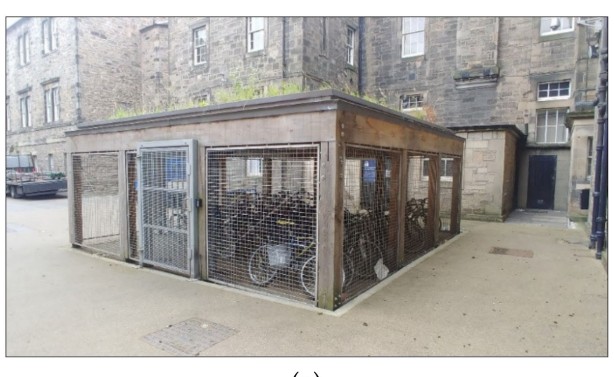
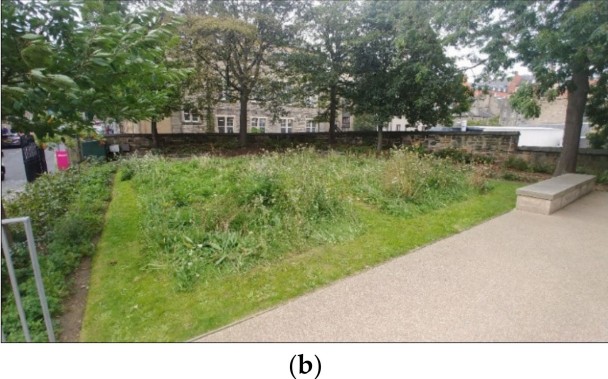

(**a**)            (**b**)

**Figure 10.** Landscaping in the ECCI's grounds; (**a**) Green roof over bike storage; (**b**) Wild grass meadow planting in garden spaces; author's photographs.

Water efficiency ensures reduced demand and, therefore, reduced environmental impact cause by wastewater processing. The ECCI had ambitions to install rainwater harvesting for greywater recycling in toilet cisterns. However, protected remains from the historic abbey were discovered during the excavation of the planned storage tanks making this unviable.

Transport, Waste, & Pollution

Regarding transport, BREEAM allocates credits for proximity to public networks and consideration for sustainable private transport (e.g., bicycles), as efforts to lessen dependence on cars reduces fossil fuel use. The ECCI's location is optimal, being within walking distance of many public transport links—which is also the case at 30 James Street—whilst on-site bicycle storage helps promote passive transportation and public exercise.

Waste and pollution can also have implications on greenhouse gas emissions through the deconstruction and disposal of materials at the end of a buildings life, although this can be hard to quantify. The design of the ECCI principally addressed these issues by relying on passive design over mechanical systems and in its sensible material palette of timber, steel, and stone, all of which can be reused or reconstituted.

## 3. Energy, Carbon & Cost Analysis

### 3.1. Sustainable Development Approach or 30 James Street

Currently, 30 James Street suffers from high energy consumption and costly bills, despite reaching an acceptable performance standard for Part L. Analysing the case studies has revealed several potential carbon saving solutions, outlined in Figure 11, that could be appropriate carbon saving methods which could improve this performance.

| Passive Design | – Update existing window units with double glazing for improved thermal performance. <br> – Reinstate window shutters to improve window thermal performance. <br> – Draught exclusion to reduce heat loss through envelope. |
|---|---|
| Low-Carbon Technology | – LEDs to improve lighting efficiency. <br> – Heat recovery system: utilising heat exchangers to reduce waste in central heating. <br> – Air source heat pumps to supplement central heating & hot water supply. |
| Renewable Energies | – Transparent (thin film) solar PVs on proposed southern roof terrace. <br> – Micro-CHP plant to provide electricity and improve hot water efficiency. |
| Additional Considerations | – In-use energy monitor and display. <br> – Greywater harvesting to improve water efficiency. |

**Figure 11.** Summary of proposed sustainable design strategies for 30 James Street; author's work.

The updating of existing glazing and addition of draught excluders both relate to fabric performance, therefore being the only strategies currently addressed under Part L. The remaining strategies look at alternative approaches to reduce the energy use and environmental impact of buildings. The scales of these interventions can be categorised as follows:

- Scale 1—The primary focus of building carbon is operating emissions, sequestered through energy efficiency and renewable energies (on-site or off-site).
- Scale 2—Including embodied carbon: the harvesting, manufacture, and transport of materials are considered, in addition to building construction and demolition. These issues are addressed through sensible material sourcing, efficient construction, and carbon offsetting—through renewables or third-party involvement.
- Scale 3—Including indirect emissions, this further looks at: transport, waste, water, and pollution. These topics require bespoke solutions.

Part L focusses purely on operational carbon (scale 1) (Figure 12), but, by the government's own definition, this should expand to consider embodied carbon (scale 2). At a higher level, BREEAM tries to address indirect environmental impacts as well (scale 3).

This proposes a host of considerations that vary dramatically in impact and price, but, realistically, proposed solutions must be financially viable to meet the economic dimension outlined in HM Government's definition of sustainable development.

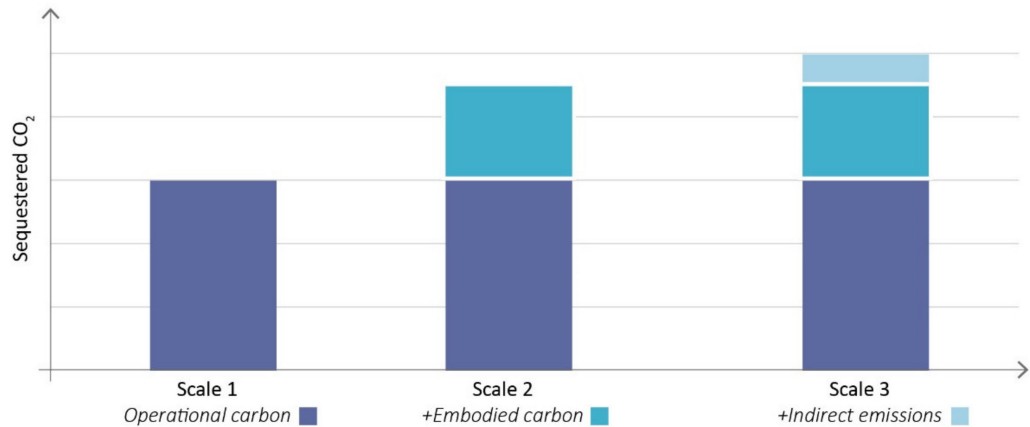

**Figure 12.** Illustrative carbons savings with varying scales of consideration; author's work.

### 3.2. Energy & Carbon Calculations

### 3.2.1. Energy & Carbon Use at 30 James Street

The light-touch of 30 James Street's retrofit added little mass of new materials, and thus accrued little embodied carbon. Without an understanding of construction details and quantities we cannot accurately calculate the embodied carbon, therefore it will be assumed negligible in comparison to lifetime emissions from operation. Referencing plans from Signature Living's Design and Access report, the floor area of 30 James Street is approximately 5000 m$^2$ [30]. Though the exact energy use is not available, multiplying figures from the CIBSE TM46 Benchmarks by the floor area can provide working approximations (Figure 13) [31].

|  | Electricity demand (kWh) | Fossil thermal (gas) demand (kWh) | Electricity Carbon (kgCO$_2$) | Fossil thermal carbon (kgCO$_2$) | Total carbon (kgCO$_2$) |
|---|---|---|---|---|---|
| **Benchmarks** | 105/m$^2$ | 330/m$^2$ | 57.8/m$^2$ | 62.7/m$^2$ | - |
| Total - 30 James Street | 525'000 | 1'650'000 | 289'000 | 313'500 | 602'500 |

**Figure 13.** Current energy and Carbon calculations for 30 James Street; author's work.

The calculated figures divide carbon emissions between fossil thermals (natural gas heating) and electricity. To give this context, it is useful to understand the CO$_2$ emissions breakdown in the building by sector. Adjusting the measured performance of a mixed-use case study allows estimates for 30 James Street to be made (Figure 14) [32]. The adjustments are: small power (TVs and appliances) is halved due to the function of 30 James Street as a hotel; heating is doubled, to account for poor thermal performance; and hot water is increased 50%, acknowledging the basement spa.

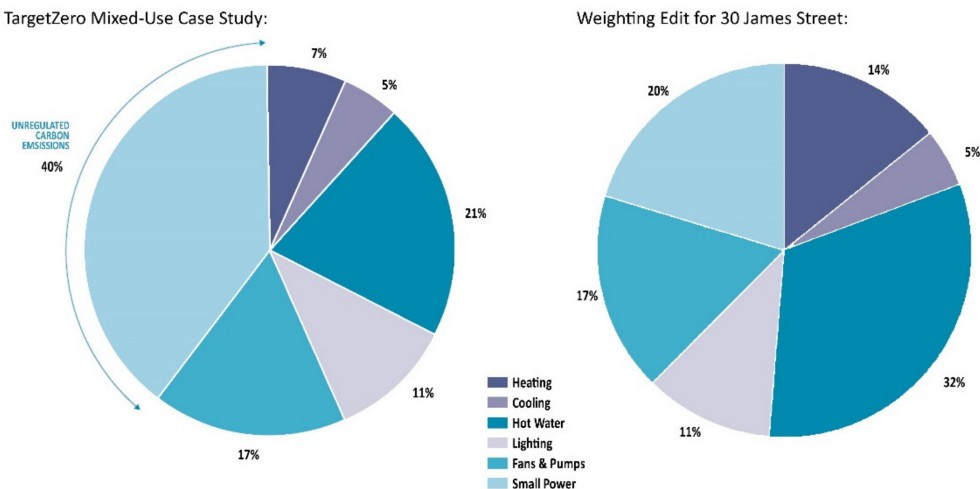

**Figure 14.** Percentage $CO_2$ emissions by sector for a mixed-use case study (**left**) and 30 James Street (**right**); author's work.

### 3.2.2. Energy & Carbon Savings of Sustainable Development Strategies

The savings in Table 1 have been calculated primarily using figures from the Historic Soho report, but adapted to suit the scale of 30 James Street. Figures for LEDs, heat recovery, heat pumps, and renewables have been separately modelled on efficiency ratings, within their target areas, taken from concurrent product data. A full breakdown of these calculations can be found in Appendix B.

**Table 1.** Energy & carbon calculations for sustainable development strategies at 30 James Street; author's work.

| Strategy | Target Area (Associated Energy %) | Annual Energy Savings (kWh) | Annual Carbon Savings (kgCO$_2$) | Total Energy/Carbon Savings |
|---|---|---|---|---|
| Update double Glazing | Heating (14%) and cooling (5%) | 19,783 | 5480 | 0.9% |
| Window shutters | Heating (14%) and cooling (5%) | 17,696 | 4902 | 0.8% |
| Draught Proofing | Heating (14%) and cooling (5%) | 46,704 | 12,938 | 2.1% |
| LEDs | Lighting (11%) | 51,975 | 28,611 | 9.9% |
| Heat recovery system | Heating (14%), hot water (32%) | 217,800 | 60,333 | 10.0% |
| Air source heat pumps | Heating (14%), and cooling (5%) | 188,100 | 52,106 | 8.6% |
| In-use energy display | General (100%) | 108,750 | 30,125 | 5.0% |
| Greywater harvesting | - | - | 3.6 | 0.0% |

The impact of passive systems is limited to windows and openings, meaning overall savings are little. Contrastingly, low-carbon technologies create significant savings overall, with the biggest reductions targeting heating and hot water systems—which constitute nearly half of existing energy consumption. In this instance, the poor thermal performance of 30 James Street causes greater savings from low-carbon technology than in better insulated buildings. Furthermore, the efficiency of LED technology versus halogen is significant, with the potential to drastically reduce overall energy consumption by 9.9%.

Considering its area, the energy production of solar glazing is low and contributes little to overall building savings (Table 2). This is potentially due to a lower efficiency of thin-film technology, as opposed to typical solar PVs [27], but also because of the large electricity demand at 30 James Street. At any rate, the savings are not significant enough to

justify it as a viable solution. Contrastingly, a biogas micro-CHP unit could supplement the entire energy and heating demand of the building and be small enough to fit into an internal plant room.

**Table 2.** Energy & carbon calculations for renewable energy generation at 30 James Street; authors' work.

| Renewable Energy | Size of Technology (m²) | Annual Energy Production (kWh) | Annual Carbon Savings (kgCO₂) | Total Energy/Carbon Savings |
|---|---|---|---|---|
| Thin-film solar glazing | 116.4 | 12,236 | 3390 | 0.6% |
| Micro-CHP (50 kW) | Small (>10) | 1,814,571 | 502,657 | 83.4% |
| Micro-CHP (80 kW) | Small (>10) | 2,175,000 | 602,500 | 100.0% |

*3.3. Quantifying Savings for 30 James Street*

Not all the tested sustainable development strategies will work in tandem so two compiled solutions are being proposed (Figure 15). Solution 1 combines passive design (updated glazing and draught proofing) with low-carbon technology (LEDs, heat pumps, heat recovery, and energy monitoring). These are well tested technologies that could presumably be implemented with few issues but would save an estimated 36.6% of energy demand and carbon emissions at 30 James Street. Comparing low-carbon technologies and passive design systems, fabric performance measures are not effective at 30 James Street and alternative technologies, such as heat exchangers, bear much greater success.

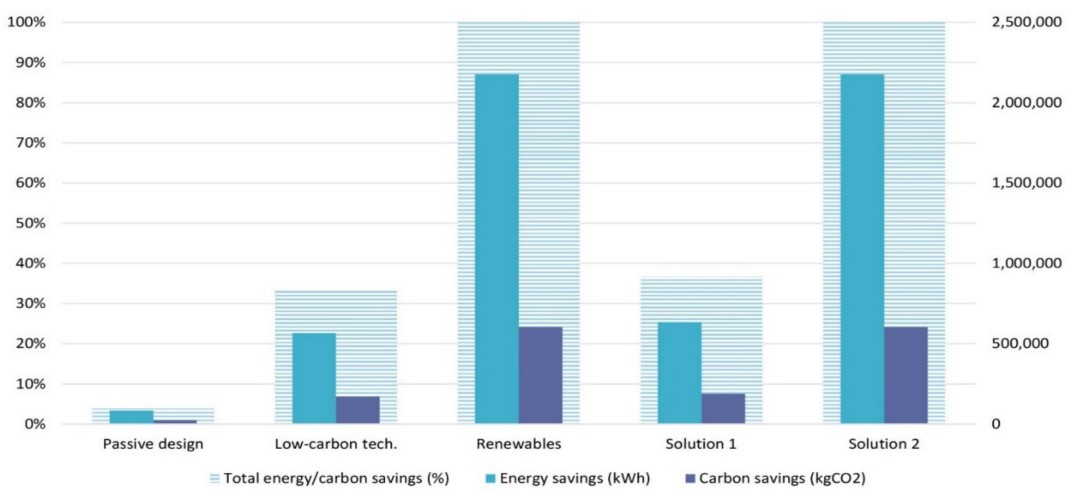

**Figure 15.** Diagram showing energy and carbon savings for proposed solutions at 30 James Street; author's work.

Solution 2 further incorporates on-site renewables by replacing heat pumps and heat recovery with a micro-CHP generator. Again, CHPs are a well-tested technology proposing little risk, and can be installed in an existing internal plant room. Combining low-carbon technologies and passive design with a CHP reduces the building's energy demand enough to require a smaller, more efficient unit (50 kW) than would otherwise be required. This could potentially make 30 James Street independent of grid electricity, and, moreover, could be switched to a renewable biogas supply, entirely negating the operational carbon of the site. (Scale 1). Excess electricity generation could further begin to compensate for the undetermined embodied carbon of the retrofit (Scale 2).

## 4. Finance & Feasibility

*4.1. Financial Implications of Sustainable Development*

To give credibility to the sustainable design strategies proposed in Section 3.3, their finances have been considered in Table 3. Again, figures from the Historic Soho report have been referenced, although the payback time has been modified to correspond with

modern energy prices [33,34]. A traffic light rating has been assigned to each strategy, illustrating the overall viability of the option with finance considered; green (viable), amber (potentially viable), and red (not viable).

**Table 3.** Cost calculations of sustainable development strategies at 30 James Street; author's work.

| Strategy | Energy Savings (kWh) | Carbon Savings (kgCO$_2$/yr) | Total Savings | Upfront Costs (%) * | Cost of Annual Savings | Payback Time (Years) | Viability |
|---|---|---|---|---|---|---|---|
| Update double glazing | 19,783 | 5480 | 0.9% | £80,000 (1.6%) | £554 | 144 | |
| Window shutters | 17,696 | 4902 | 0.8% | £44,000 (0.9%) | £495 | 89 | |
| Draught proofing | 46,704 | 12,938 | 2.1% | £46,000 (0.9%) | £1308 | 35 | |
| LEDs ** | 51,975 | 28,611 | 9.9% | £20,000 (0.4%) | £6445 | 3.1 | |
| Heat recovery system | 217,800 | 60,333 | 10.0% | £16,000 (0.3%) | £6098 | 2.6 | |
| Air source heat pumps | 188,100 | 52,106 | 8.6% | £16,000 (0.4%) | £5267 | 3.0 | |
| In-use energy display | 108,750 | 30,125 | 5.0% | £1,000 (0.0%) | £5565 | >1 | |
| Greywater harvesting | - | 3.6 | 0.0% | £2,250 (0.0%) | £0 | - | |
| Thin-film solar glazing | 12,236 | 3390 | 0.6% | £29,100 (0.6%) | £1517 | 19 | |
| Micro-CHP (50 kW) | 1,814,571 | 502,657 | 83.4% | £123,000 (2.5%) | £31,857 | 3.9 | |
| Micro-CHP (80 kW) | 2,175,000 | 602,500 | 100.0% | £160,000 (3.2%) | £10,122 | 16 | |

Note: * Upfront costs also shown as a percentage of initial retrofit cost (£5 million); ** Assumption of 4000 bulbs required at 30 James Street with estimated £5 price per bulb; green (viable), amber (potentially viable), and red (not viable).

When viewed through from a financial perspective some of the design solutions become unviable. For instance, the relative cost of updating double glazing against installing shutters is almost double, which is not justified by their similar carbon savings and makes it an unviable option. The low return on energy savings of the shutters, however, means they will likely never pay back their upfront costs and therefore are also undesirable. Similarly, solar panels are expected to last 20 years, so a payback time of 17 years here, not including maintenance, yields little profit on investment. Although there is still a positive environmental impact, we cannot expect private owners to invest in systems that offer them no financial benefit. Antithetically, greywater recycling also provides very low return on savings, painting it as economically impractical, but this figure is not representative as such systems can pay for themselves through water bill savings [35], revealing another flaw in the current energy focus of Part L.

We see that low carbon technology exhibits the lowest upfront costs, but also some of the largest energy and carbon savings. These systems have the shortest payback times, giving them the highest proportional energy savings and showing that there are suitable alternatives to Part L's fabric-first approach. Interestingly, building fabric improvement is noticeably more expensive but, collectively, show low enough energy reductions that the most effective solution—draught proofing, which typically comes with 20-year guarantees [36]—will still yield no profits in its lifespan. An indoor micro-CHP (50 kW) could replace the existing plant facilities at 30 James Street and pay for itself within several years. As biogas supplies and green electricity become more commonly available, the return on investment also has the potential to grow. As such, a micro-CHP could be a highly successful intervention at 30 James Street, with its only initial concern being large upfront costs.

*4.2. Feasible Approaches to Sustainable Development at 30 James Street*

Looking again at the proposed solutions for 30 James Street (Figure 16), the high costs and long payback periods of all passive design strategies render them unviable. This is acknowledged in solutions 3 and 4, corresponding to solutions 1 and 2 but with passive

design measures removed. The retrofit of 30 James Street in 2013 was given funding of £5 million. To install the strategies in Solution 3 would cost an estimated £54,000, equating to 1.1% additional spending but potentially saving c.33.6% of annual energy consumption. That translates to £23,375 in annual cost savings, therefore paying for all sustainable design measures in a little over 2 years. Theoretically, this solution is cheap, provides significant environmental and economic benefits, and could be repeated on existing buildings across the UK to similar success.

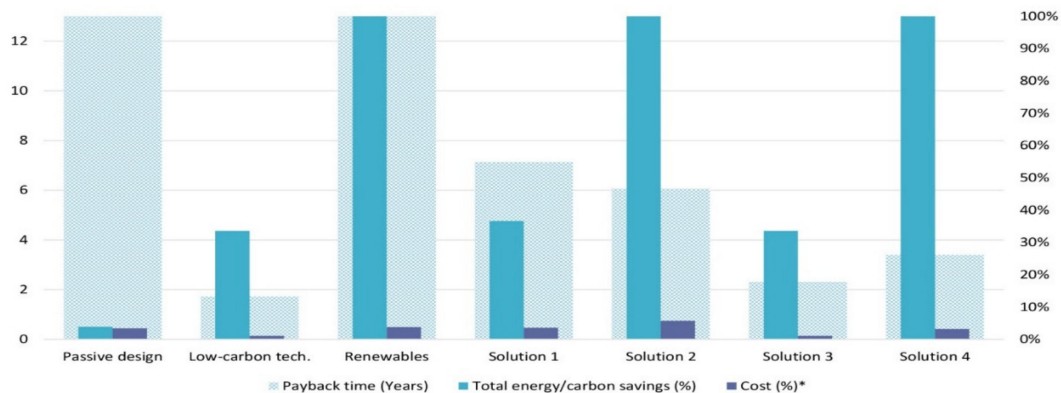

**Figure 16.** Diagram showing costs and savings for proposed solutions at 30 James Street; author's work.

Solution 4 has the potential to negate all building carbon through a renewable biogas supply. Accounting for the cost of the biogas, annual savings still allow a payback time of ca. 3.4 years. The issue comes with higher upfront costs, although HM Government do provide several incentives to subsidise on-site renewable energy generation. The Renewable Heat Incentive (RHI), and Feed-In Tariff (FIT) reward localised heat and energy generation, although the FIT has now been closed to new members [37]. Yet what serves as better incentive was the Green Deal, which had the government cover upfront costs of procurement and installation, to be repaid through energy bill savings [38]. The deal collapsed due to poor consumer and investor appeal and has only since been replaced by the Green Homes Grant for domestic works and loans from private enterprises [39]. However, this could have been mitigated with better advertisement in the form of sustainable design guidance and heightened performance standards, as promoting the benefits of improved energy efficiency has been shown to accelerate consumer uptake [40].

## 5. Discussion

The primary goal of HM Government's 2050 targets is to address the UK's environmental impact. Through Part L, this has started to be tackled in the energy use of buildings, although the study of 30 James Street has proven that a fabric-first approach is not an effective solution on all existing buildings.

Whilst there is no definitive number of buildings in the UK, we have discussed that the construction industry constitutes 45–50% Britain's carbon footprint. Under Part L, an existing building is defined as any refurbishment or fit-out project working with existing building fabric. For the sake of argument, however, we will assume that any building constructed before the year 2000 has adhered to a lower sustainability standard and will be categorised as 'existing'. As discussed in the Introduction, this applies to an estimated 88.7% of English homes which would fall under Part L's existing domestic standard. Assuming a similar figure is true for all buildings in the UK, including non-domestic, then most buildings will be subject to the existing building standards of Part L, despite the new build standards being significantly more stringent.

If we could repeat solution 3—a 33.6% energy and carbon reduction—across all existing buildings, we could save c.29.8% of construction industry carbon. This would diminish the total carbon emissions of the UK by c.13.4% (assuming 45% contributed by

the construction industry), equal to c.48.8 billion $kgCO_2$ [41]. Though this calculation is obviously flawed, due to figures being heavily based on estimates, if only a third of the estimated savings were attained it would still contribute a c.7.8% reduction in building industry and c.4.5% of overall UK carbon emissions—equivalent to ~16.3 billion $kgCO_2$. This is a sizeable reduction yet comes with relatively small upfront costs and goes on to provide returns on investment with several years. If on-site renewables were also considered, many buildings could further become self-sufficient, at least in part, therefore reducing demand on the grid, as per Part L's current trajectory. That said, the retrofit of 30 James Street was a large scheme that could more easily absorb expensive upfront costs for CHPs and other such technologies. Though smaller units would be required on smaller scale works, the cost of CHPs is far higher than alterative boilers and, realistically, many smaller building projects might require loans to fund procurement. Although longer established renewables are more affordable, such as solar PVs, a new green deal for non-domestic works would be pivotal in alleviating the burden of capital costs for on-site renewables.

### 5.1. Immediate Revisions to Part L

Contrasting the current fabric-first approach to sustainable design exhibited by Part L, a superior strategy for reducing the energy demand and carbon emissions of existing buildings would incorporate a wide variety of the sustainable design technologies available on the modern market. Figure 17 shows a full breakdown of these technologies, highlighting considerations currently addressed by Part L in red. As shown in the analysis of 30 James Street, strategies in the categories of low-carbon technology and on-site renewables show great promise in improving the efficiencies of lighting, heating, and hot water supply in existing buildings. As such, it is reasonable to assume that similar strategies have potential to show similar effectiveness on other retrofits across the UK, providing strong evidence that the Approved Documents L1b and L2b require immediate revisions.

This is not to suggest that fabric performance is of no benefit to retrofits; as identified in the case of the ECCI, high envelope performance can be a significant factor in reducing carbon emissions and achieving low-environmental impact, however it is not always a viable option. For this reason, providing multiple paths towards achieving good sustainable performance would improve the chances of existing buildings to significantly reduce their carbon footprints and environment impacts. This could operate like BREEAM, allowing designers to focus on a mixture of approaches and could transfer from energy performance benchmarks to attaining carbon emission benchmarks. For example, switching to a standard that measured carbon emissions per unit area over time (e.g., $CO_2/m^2$ per hour) would see the positive impact of on-site renewables counteract poor thermal performance in building fabric, though this requires further investigation to properly understand and implement.

Diverging from the focus on energy efficiency, Part L must also grow to address all construction industry carbon if it is to meet the zero-carbon target. This means that the lifetime embodied carbon of buildings must be considered. Although 30 James Street did not afford an in-depth investigation into this matter, the ECCI shows that considering local materials, landscaping, and on-site renewables are all viable solutions in addressing these emissions. Subsequent research into finance might allow these ideas to also be implemented as guidance into Part L.

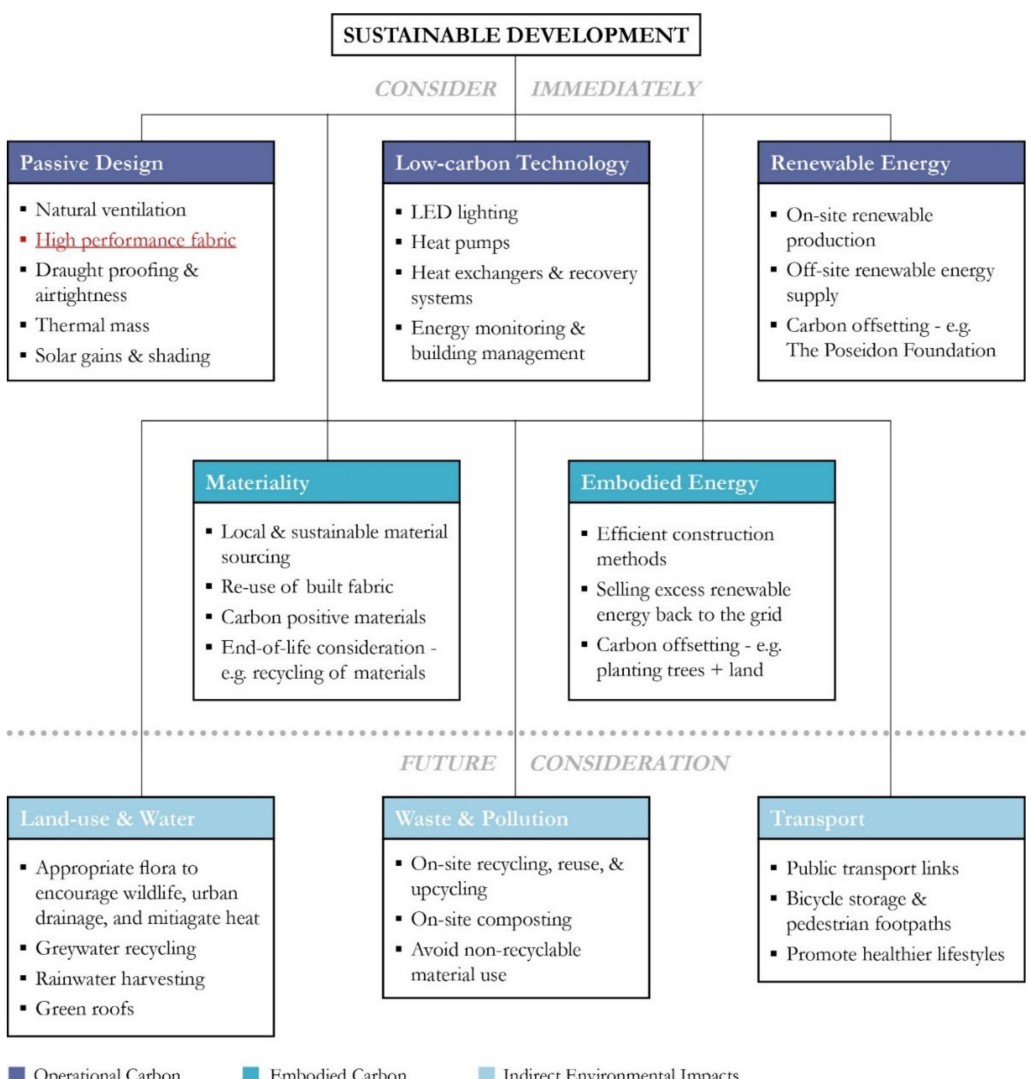

**Figure 17.** Initial considerations in developing a framework of revisions for Approved Documents L1b and L2b; author's work.

### 5.2. Future Considerations for Part L

A misconception that has been perpetuated by the energy focus of Part L is the idea that carbon emissions are the sole perpetrator of the UK's environmental impact. The carbon benefits of rainwater harvesting and greywater recycling have been briefly mentioned, although, because they do not factor into the existing sustainable design model of Part L, it is difficult to appreciate their potential. Similarly, we know that land-use is directly linked to biodiversity and air quality, amongst other benefits, but it does not directly impact energy or carbon savings. Therefore, there is an argument to further expand the focus of Part L to include indirect environmental impacts and emissions, as these concerns still fall under the bracket of "protecting our environment" outlined in HM Government's sustainable development definition.

### 6. Conclusions

#### 6.1. Research Implications

The study sought to analyse the effectiveness of HM Government's fabric-first approach to sustainable development when addressing existing building projects and to further identify alternative methods that showed promise in reducing the environmental impact of the construction industry. Using two case studies, a variety of methods towards

achieving reduced energy demand and carbon emissions have been identified and tested, making it evident that the existing standards of Approved Documents L1b and L2b are too narrowly focussed and should be immediately revised to reflect the full range of modern sustainable development solutions available.

Although a fabric-first approach has been successfully implemented in existing buildings, it is not always an effective solution. As seen at 30 James Street, where building fabric is problematic and protected, envelope performance can be severely restricted by external factors, rendering the guidance of Approved Document L too narrowly focussed. The analysis of 30 James Street has shown that alternative, cost-effective sustainable design approaches in the forms of low-carbon technology and on-site renewable energy generation can be equally effective in reducing the energy demand and carbon emissions of operating buildings. Expanding the guidance of Approved Documents L1b and L2b to reflect this would address the challenges faced in attaining good envelope performance by many existing buildings and would further provide them more flexibility in choosing pathways towards becoming sustainable developments, reflecting their complexities and nuances.

Embodied carbon also constitutes a significant portion of construction industry $CO_2$ and Approved Document L must immediately adapt to consider this if HM Government's ambitions of becoming zero-carbon are to be realised. As observed in the study of the ECCI, many factors contribute to embodied carbon, from materiality to construction techniques. These considerations could be implemented as guidance in the documents to mitigate carbon intensive construction methods, such as by promoting the use of timber. Furthermore, on-site renewables provide an opportunity to compensate for embodied carbon whilst also decreasing dependence on grid electricity, therefore helping the government transfer to all-renewable energy generation.

Additionally, it is reasonable to conclude that alternative methods of addressing the construction industry's environmental impact exist through means of landscaping, sustainable transport, waste and water management, and building management. These are all criteria found already in other standards, like BREEAM, that align with HM Government's definition of sustainable development by "maximising wellbeing and protecting our environment". Therefore there is scope to research and implement sustainable design methods under these categories as guidance in Approved Document L in future.

### 6.2. Future Areas of Investigation

The study has identified viable sustainable development methods that pose potential in updating the existing building standards of Approved Document L, however what has not been broached in detail is their implementation. One proposal contributed in the discussion would be moving from energy measuring methods towards carbon measuring, for example with $CO_2$ emission benchmarks per unit area. Another option would mirror the BREEAM approach, assigning credits for the efforts of various sustainable design solutions. Both concepts present opportunities and constraints, so the most appropriate assessment methods would need to be determined for the revised standards.

Additionally, though the study has concentrated on existing buildings, the proposed revisions might also reap benefits on the new build standards. In particular, low-carbon technology and on-site renewables might be equally effective in new builds, especially regarding technologies like LED lighting. Further studying the potential of these solutions could justify their addition to Approved Documents L1a and L2a. Moreover, a wider range of sustainable development solutions and guidance may advocate a universal standard in Approved Document L between new and existing building projects. All projects could then attain comparable sustainability levels through varying means.

As touched on in Section 5.2, there are also several additional considerations pertaining to maximising wellbeing and protecting our environment that are not currently considered under Part L. Land-use, water management, and materiality, amongst others, are considerations known to have positive impacts on user wellbeing and environmental impact. As these criteria are already included in BREEAM, their merits should be

assessed and potentially included in the Approved Documents as well. Furthermore, clarity needs to be given to HM Government's definition of "protecting our environment" in their sustainable development definition, as $CO_2$ emissions, which are currently the absolute focus of the standards, are not the only significant environmental impact of the construction industry.

**Author Contributions:** Conceptualization, A.W. and S.F.; methodology, A.W. and S.F.; formal analysis, A.W.; investigation, A.W.; resources, A.W. and S.F.; data curation, A.W.; writing—original draft preparation, A.W.; writing—review and editing, A.W. and S.F.; visualization, A.W. and S.F.; supervision, S.F. All authors have read and agreed to the published version of the manuscript.

**Funding:** This research received no external funding.

**Institutional Review Board Statement:** Not applicable.

**Informed Consent Statement:** Not applicable.

**Data Availability Statement:** Not applicable.

**Acknowledgments:** This study was completed in partial completion of the course of Master of Architecture (March) at the University of Liverpool on behalf of the author, Andrew Williamson. We give our appreciation to the employees of Signature Living, Malcolm Fraser Architects, the 30 James Street Hotel, and the Edinburgh Centre for Carbon Innovation, who indulged our research and made this study possible.

**Conflicts of Interest:** The authors declare no conflict of interest.

## Appendix A

*Definitions & Acronyms*

- Building fabric—components of a building's structure: e.g., walls, floors, roofs.
- Carbon footprint—The total $CO_2$ released by a building through its lifetime, including: construction, operation, and demolition.
- CIBSE—The Chartered Institute of Building Services Engineers
- Envelope—The components of building fabric that separate internal and external space: e.g., exterior walls, roofs, and ground floors/basements.
- Fabric-first approach—relating to sustainable design, this approach looks to improve the thermal performance of building fabric to lower energy consumption from heating/cooling.
- HM Government—Her Majesty's Government of Great Britain
- Operational carbon/operational emissions—$CO_2$ emissions caused by the daily operations of buildings (e.g., heating, lighting, and ventilation).
- Renewable energy/renewables—Energy generation sources which do not deplete natural resources (e.g., solar, wind, biomass).
- Retrofitting—The replacement or alteration of existing buildings/built fabric and services to allow for new functions. Within the context of the paper, this mainly refers to improving the low-carbon performance of existing buildings.
- Thermal bridge—Relating to 'Thermal transmittance', a bridge (verb. bridging) is an element with a higher transmittance than its surroundings, causing a path of less resistance that increases heat loss.
- Thermal transmittance—Otherwise referred to as a U-value. This is the rate at which heat passes through an object, such as through the envelope of a building. A lower transmittance implies more heat retention, and therefore lower energy required to maintain thermal comfort.

## Appendix B

*30 James Street figures for calculations:*

- Net Floor Area = 5000m²
- ~200 windows at 30 James Street (based on Signature Living elevations)
- Current annual energy consumption = 2,175,000 kWh
- Current annual $CO_2$ emissions = 602,500 $kgCO_2$

---

### UPDATE DOUBLE GLAZING
*Costs:*

- Avg. cost = £400/window

  200 x £400 = £80,000 upfront price

  (80,000/5,000,000) x 100 = 1.6% cost of original retrofit

  19,783 kWh x 2.8p = £554 annual energy bill savings

  £80,000 / £554 = 144 year payback period

*Energy & carbon savings:*

- Avg. carbon saving = 27.4 $kgCO_2$/window

  200 x 27.4 $kgCO_2$ = 5480 $kgCO_2$ overall reduction

  (5,480/602,500) x 100 = 0.9% total savings

  0.9% x 2,175,000 kWh = 19,783 kWh energy savings

---

### WINDOW SHUTTERS
*Costs:*

- Avg. cost = £220/window

  200 x £220 = £44,000 upfront price

  (44,000/5,000,000) x 100 = 0.9% cost of original retrofit

  19,783 kWh x 2.8p = £495 annual energy bill savings

  £80,000 / £495 = 89 year payback period

*Energy & carbon savings:*

- Avg. carbon saving = 24.51 $kgCO_2$/window

  200 x 24.51 $kgCO_2$ = 4902 $kgCO_2$ overall reduction

  (4902/602,500) x 100 = 0.8% total savings

  0.8% x 2,175,000 kWh = 17,696 kWh energy savings

---

### DRAUGHT PROOFING
*Costs:*

- Avg. cost = £40/m²
- Area of envelope openings ~1150m²

  1150m² x £40 = £46,000 upfront price

  (46,000/5,000,000) x 100 = 0.9% cost of original retrofit

  46,704 kWh x 2.8p = £1308 annual energy bill savings

  £46,000 / £1308 = 35 year payback period

*Energy & carbon savings:*

- Avg. carbon saving = 11.25 $kgCO_2$/m²

  1150m² x 11.25 = 12,938 $kgCO_2$ overall reduction

  (12,938/602,500) x 100 = 2.1% total savings

  2.1% x 2,175,000 kWh = 46,704 kWh energy savings

---

**Figure A1.** Cost, energy, and carbon calculations for sustainable development strategies at 30 James Street [Page 1].

**LED LIGHTING**

*Costs:*

- Avg. cost = £5/bulb
- Estimated 4000 fittings throughout the hotel

    4000 x £5 = £20,000 upfront price

    (20,000/5,000,000) x 100 = 0.4% cost of original retrofit

    51,975 kWh x 12.4p = £6445 annual energy bill savings

    £20,000 / £6445 = 3.1 year payback period

*Energy & carbon savings:*

- Energy use (compared to halogen) = 10% (90% reduction)
- Lighting equates to 11% of energy use at 30 James Street

    90% x 11% = 9.9% total savings

    9.9% x 602,500 $kgCO_2$ = 28,611 $kgCO_2$ overall reduction

    9.9% x 2,175,000 kWh = 51,975 kWh energy savings

---

**HEAT RECOVERY SYSTEM**

*Costs:*

- Avg. cost of £8000 for HVAC recovery and condensing gas-to-water boiler economiser.

    £8000 + £8000 = £16,000 upfront price

    (16,000/5,000,000) x 100 = 0.3% cost of original retrofit

    217,800 kWh x 2.8p = £6098 annual energy bill savings

    £16,000 / £6098 = 2.6 year payback period

*Energy & carbon savings:*

- Avg. heat recovery efficiency of 60% in HVAC systems and 15% for condensing gas-to-water boiler economiser.
- Hot water 32%, Heating 14%, and cooling 5% of energy use at 30 James Street

    ((32% x 15%) + (19% x 60%)) x 1,650,000 = 217,800 kWh energy reduction

    (217,800/2,175,000) x 100 = 10.0% total savings

    10.0% x 602,500 $kgCO_2$ = 60,333 $kgCO_2$ energy savings

---

**AIR SOURCE HEAT PUMPS**

*Costs:*

- Avg. ASHP cost of £8000

    £10,000 + £10,000 = £16,000 upfront price

    (16,000/5,000,000) x 100 = 0.3% cost of original retrofit

    188,100 kWh x 2.8p = £5267 annual energy bill savings

    £16,000 / £5267 = 3.0 year payback period

*Energy & carbon savings:*

- Avg. energy efficiency of 60% in air-to-air systems

    (0.19% x 2,175,00) / (365x24) = ~47kW

- Estimate 2 small units to supply all heating and cooling to 30 James Street (19%)

    (19% x 60%) x 1,650,000kWh = 188,100 kWh energy reduction

    (188,100/2,175,000) x 100 = 8.6% total savings

    8.6% x 602,500 $kgCO_2$ = 52,106 $kgCO_2$ energy savings

---

**Figure A2.** Cost, energy, and carbon calculations for sustainable development strategies at 30 James Street [Page 2].

**IN-USE ENERGY MONITOR**

*Costs:*

- Estimated £2000 for display and monitors (for rooms & systems).

    (2000/5,000,000) x 100 = 0.0% cost of original retrofit

    (5% x 1,650,000 kWh x 2.8p) + (5% x 525,000 x 12.4p) = £5565 annual energy bill savings

    £2000 / £5565 = >1 year payback period

*Energy & carbon savings:*

- Avg. overall energy use streamlined by 5%

    5% x 2,175,000 = 108,750 kWh energy reduction

    5% x 602,500 $kgCO_2$ = 30,125 $kgCO_2$ energy savings

---

**THIN FILM SOLAR PVs**

*Costs:*

- Approximate cost of £250/m².

    £250 x 116.4 m² = £29,100 upfront price

    (29,100/5,000,000) x 100 = 0.6% cost of original retrofit

    12,236 kWh x 12.4p = £1517 annual energy bill savings

    £29,100 / £1517 = 19 year payback period

*Energy & carbon savings:*

- Maximum 72W output energy.
- Average sun lighting in the UK = 4 hours/day
- 116.4m² glazed wall and roof area on terrace

    4 hours x 365 x 0.072kW x 116.4 = 12,236 kWh energy production

    (12,236/2,175,000) x 100 = 0.6% total savings

    0.6% x 602,500 $kgCO_2$ = 3390 $kgCO_2$ energy savings

---

**MICRO-CHP (50kW)**

*Costs:*

- Installation cost estimated at £50,000
- Unit cost gathered from UK Tedom CHP supplier
- Approximate cost of biogas = 5p/kWh

    ~£73,000 + £50,000 = £123,000 upfront price

    (123,000/5,000,000) x 100 = 2.5% cost of original retrofit

    (438,000 x 12.4p)+((889,140+ 592,760) x 2.8p = £95,805 annual energy bill savings

    (146kW x 24 x 365) x 5p = £63,948 cost of biogas supply annually

    £95,805 - £63,948 = £31,857 net annual savings

    £123,000 / £31,857 = 3.9 year payback period

*Energy & carbon savings:*

- Maximum 51kW electricity output & 101.5kW heat output (146kW biogas supply)
- CHP boilers are 60% more efficient than traditional HVAC systems

    24 hours x 365 days x 50kW = 438,000 kWh electricity production

    24 hours x 365 days x 101.5kW = 889,140 kWh heating production

    (889,140/0.6) - 889,140 = 592,760 kWh gas saved for heating

    83.4% x 2,175,000 kWh = 1,814,571 kWh energy saved

    83.4% x 602,500 $kgCO_2$ = 502,657 $kgCO_2$ saved

---

**Figure A3.** Cost, energy, and carbon calculations for sustainable development strategies at 30 James Street [Page 3].

**MICRO-CHP (80kW)**

*Costs:*

- Installation cost estimated at £50,000
- Unit cost gathered from UK Tedom CHP supplier
- Approximate cost of biogas = 5p/kWh

  ~£110,000 + £50,000 = £160,000 upfront price

  (160,000/5,000,000) x 100 = 3.2% cost of original retrofit

  (525,000 x 12.4p)+(1,650,000x 2.8p) = £111,300 annual energy bill savings

  (231kW x 24 x 365) x 5p = £101,178 cost of biogas supply annually

  £95,805 - £63,948 = £10,122 net annual savings

  £160,000 / £10,122 = 16 year payback period

*Energy & carbon savings:*

- Maximum 81kW electricity output & 126kW heat output (231kW biogas supply)
- CHP boilers are 60% more efficient than traditional HVAC systems

  24 hours x 365 days x 81kW = 709,560 kWh electricity production

  24 hours x 365 days x 126kW = 1,103,760 kWh heating production

  (1,103,760/0.6) - 1,103,760 = 735,840 kWh gas saved for heating

  *Excess heat & gas can be sold back to the grid for profit*

**Figure A4.** Cost, energy, and carbon calculations for sustainable development strategies at 30 James Street [Page 4].

| Strategy | Solution 1 | Solution 2 | Solution 3 | Solution 4 |
|---|---|---|---|---|
| Update double Glazing | ■ | ■ | | |
| Window shutters | | | | |
| Draught Proofing | ■ | ■ | | |
| LEDs | ■ | ■ | ■ | ■ |
| Heat recovery system | ■ | ■ | | ■ |
| Air source heat pumps | ■ | | ■ | |
| In-use energy monitor | ■ | ■ | | ■ |
| Greywater harvesting | | | | |
| Solar PVs | | | | |
| Micro-CHP (50kW) | | ■ | | ■ |
| Micro-CHP (80kW) | | | | |

**Figure A5.** Table showing sustainable development strategies included in solutions proposed at 30 James Street.

**SOLUTION 1**

- Updated double glazing, Draught proofing, LEDs, Heat Recovery, ASHPs, and an in-use energy display.
- Total energy saved = 633,111 kWh (36.6%)
- Total carbon sequestered = 189,592 kgCO$_2$

*Costs:*

- Upfront cost = £180,000 (3.6% additional cost of original retrofit)
- Total savings = £25,237 (22.7% of est. current annual energy bill)
- Payback period:

$$£180,000/£25,237 = 7.1 \text{ years}$$

---

**SOLUTION 2**

- Updated double glazing, Draught proofing, LEDs, Heat Recovery, an in-use energy display, and a micro-CHP generator (50kW).
- Total energy saved = 2,175,000 kWh (100%)
- Total carbon sequestered = 602,500 kgCO$_2$

*Costs:*

- Upfront cost = £287,000 (5.7% additional cost of original retrofit)
- Total savings = £47,352 (42.5% of est. current annual energy bill)
- Payback period:

$$£287,000/£47,352 = 6.1 \text{ years}$$

---

**SOLUTION 3**

- LEDs, Heat Recovery, ASHPs, and an in-use energy display.
- Total energy saved = 631,025 kWh (33.6%)
- Total carbon sequestered = 189,014 kgCO$_2$

*Costs:*

- Upfront cost = £54,000 (1.1% additional cost of original retrofit)
- Total savings = £23,375 (21.0% of est. current annual energy bill)
- Payback period:

$$£54,000/£23,375 = 2.3 \text{ years}$$

---

**SOLUTION 4**

- LEDs, Heat Recovery, an in-use energy display, and a micro-CHP generator (50kW).
- Total energy saved = 2,175,000 kWh (100%)
- Total carbon sequestered = 602,500 kgCO$_2$

*Costs:*

- Upfront cost = £161,000 (3.2% additional cost of original retrofit)
- Total savings = £47,352 (42.5% of est. current annual energy bill)
- Payback period:

$$£161,000/£47,352 = 3.4 \text{ years}$$

---

**Figure A6.** Cost calculation spreadsheet for proposed sustainable development solutions at 30 James Street.

| Strategy | Energy savings (kWh) | Carbon savings (kgCO₂) | Total energy/carbon savings (%) | Cost | Cost (%)* | Cost savings | Payback time (Years) |
|---|---|---|---|---|---|---|---|
| Update double Glazing | 19,783 | 5,480 | 0.9% | £80,000 | 1.6% | £554 | 144 |
| Window shutters | 17,696 | 4,902 | 0.8% | £44,000 | 0.9% | £495 | 89 |
| Draught Proofing | 46,704 | 12,938 | 2.1% | £46,000 | 0.9% | £1,308 | 35 |
| LEDs | 51,975 | 28,611 | 9.9% | £20,000 | 0.4% | £6,445 | 3.1 |
| Heat recovery system | 217,800 | 60,333 | 10.0% | £16,000 | 0.3% | £6,098 | 2.6 |
| Air source heat pumps | 188,100 | 52,106 | 8.6% | £16,000 | 0.3% | £5,267 | 3.0 |
| In-use energy monitor | 108,750 | 30,125 | 5.0% | £2,000 | 0.0% | £5,565 | >1 |
| Greywater harvesting | - | 3.6 | 0.0% | £2,250 | 0.0% | - | - |
| Solar PVs | 12,236 | 3,390 | 0.6% | £29,100 | 0.6% | £1,517 | 19 |
| Micro-CHP (50kW)** | 1,814,571 | 502,657 | 83.4% | £123,000 | 2.5% | £31,857 | 3.9 |
| Micro-CHP (80kW)** | 2,175,000 | 602,500 | 100.0% | £160,000 | 3.2% | £10,122 | 16 |
| | | | | | | | |
| Passive design | 84,182 | 23,320 | 3.9% | £170,000 | 3.4% | £10,439 | 16.3 |
| Low-carbon tech. | 566,625 | 171,175 | 33.6% | £54,000 | 1.1% | £31,295 | 1.7 |
| Renewables | 2,175,000 | 602,500 | 100.0% | £189,100 | 3.8% | £10,122 | 18.7 |
| | | | | | | | |
| Solution 1 | 633,111 | 189,592 | 36.6% | £180,000 | 3.6% | £25,237 | 7.1 |
| Solution 2 | 2,175,000 | 602,500 | 100.0% | £287,000 | 5.7% | £47,352 | 6.1 |
| | | | | | | | |
| Solution 3 | 631,025 | 189,014 | 33.6% | £54,000 | 1.1% | £23,375 | 2.3 |
| Solution 4 | 2,175,000 | 602,500 | 100.0% | £161,000 | 3.2% | £47,352 | 3.4 |
| | | | | | | | |

| | | | | | | | |
|---|---|---|---|---|---|---|---|
| Total gas (kWh) | 1,650,000 | | Electricity cost / kWh | £0.124 | | Current annual spending | |
| Total electricity (kWh) | 525,000 | | Gas cost / kWh | £0.028 | | Gas: | £46,200 |
| Gas carbon (kgCO2) | 313,500 | | Biogas cost / kWh | £0.050 | | Electric: | £65,100 |
| Electricity carbon (kgCO2) | 289,000 | | Gross floor area (m2) | 5,000 | | Total: | £111,300 |
| Total carbon (kgCO2) | 602,500 | | | | | | |
| Total energy (kWh) | 2,175,000 | | | | | | |

*Sustainable intervention cost as a percentage of initial retrofit costs (£5million)
**A larger CHP is needed where no other energy saving measures are in place (i.e. 210kW)
***Cost savings factor in price of biogas to fuel micro-CHP

| | Electric (kWh) | Gas (kWh) | Micro-CHP (50kW) | | Micro-CHP (80kW) | |
|---|---|---|---|---|---|---|
| Solution 1 cost: | 227,211 | 405,900 | Electricity output (kWh) | 438,000 | Electricity output (kWh) | 709,560 |
| Solution 2 cost: | 525,000 | 1,650,000 | Hot water output (kWh) | 889,140 | Hot water output (kWh) | 1,103,760 |
| Solution 3 cost: | 225,125 | 405,900 | Gas energy saved (kWh) | 592,760 | Gas energy saved (kWh) | 361,680 |
| Solution 4 cost: | 525,000 | 1,650,000 | Total energy (kWh) | 1,919,900 | Total energy (kWh) | 2,175,000 |
| | | | Energy savings (not incl. fuel cost) | £95,805 | Energy savings (not incl. fuel cost) | £111,300 |
| | | | Biogas cost @ 146kW | £63,948 | Biogas cost @ 231kW | £101,178 |
| | | | | | | (+ electricity sold back to grid) |

*Costs for micro-CHP units calculated with assistance from a UK CHP supplier*

**Figure A7.** Expanded table of cost, energy, and carbon figures for sustainable development strategies and solutions at 30 James Street.

For the  Tedom Cento T210 Indoor Acoustic Canopy CHP Unit please find attached:-

- Standard Data Sheet

- General Arrangement Dimensioned Drawing.

- Typical Heat Rejection Radiator GA Drawing.

- Typical Charge-Air Cooler GA Drawing.

 In terms of price, the devil is in the detail because there are many aspects to consider, and every project is different. Typical capital cost would be i.r.o. £ 171,000 +VAT.

- The scope of supply for that price would include:

- Standard Acoustic Sound Enclosure.

- Standard 80 / 90 ΔT Operating Temperature · Top Cable Entry Switchboard

- Shentongroup Standard Switchboard Design · Ethernet Remote Monitoring Module ·

- G99 Relay (Site Interface Panel V1)

- Silencer Mounting Kit

- Charge Air Cooler radiator*

- Heat Rejection Radiator *

- Control and Supply of Heat Rejection Radiator · Bulk Oil Tanks · Bulk Oil Tank Connection Kit ·

- G99 Application Process

- Nominal Sum for Delivery

- Project Management

- Comissioning

- G99 Witness Testing

 *these need to go outside somewhere.

 What is not included in the above are things like planning permission, building regs compliance, and the installation costs.

Regarding installation, this embraces several disciplines – detailed design / electrical / controls / mechanical / gas supply / ventilation / exhaust flue etc / buildings works / civils / lift-&-shift. Every project is different, and on a unit this size a  budget of £ 50,000 for the install would be conservative.

• Yes, the original price of £171,000 doe not include VAT. In trade and industry almost every organisation claims VAT back from the government so this is irrelevant when dong cost comparisons.

• Installation – yes installation cost will be on top. As I mentioned, every project is different, and so with no accurate information about the requirement if this site, we can't even begin to cost up installation. That is why I said a budget of £50,000 for a CHP this size would be conservative.

• Cento T80 to similar spec as last time, capital cost - £109,846.18 + VAT

• Micro T50 to similar spec as last time, capital cost - £ 73,061.32 + VAT

**Figure A8.** Email liaisons with a UK-based supplier for Tedom micro-CHP units.

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
