# Peer review of "Sustainability in Heritage Buildings: Can We Improve the Sustainable Development of Existing Buildings under Approved Document L?"

_sustainability, doi:10.3390/su13073620_

Round 1

Reviewer 1 Report

The paper presents a dissertation that questions the adequacy of the standards of the Approved Document L compared to BREAAM in improving the sustainability performance of listed buildings in the UK. The approach is integrated and holistic but at the same time provides enough quantitative evidence to support its research hypotheses. The paper content is highly current, original, while documented by meticulous writing and clear aims and objectives. Literature review is comprehensive and satisfactory. The methodology is clear and well structured, the case study investigation as well is the results are sound. Minor comments on the methodology and other parts of the paper are mentioned below.

In the methodology chapter, the author mentions the rationale of selecting case study buildings. The fact that both are listed buildings is not alone adequate to support the selection of them. References to size, use, and locational attributes are needed for sounder documentation.

At the beginning of chapter four it will be useful to provide some rough drawings or sketches of the selected buildings, side by side, to realize the size of each building through a visual comparison.

Figure 17 needs a reference.

Chapter 6 refers to different solutions (proposals/ mitigation measures), but it is not explained what is the aim is of examining different solutions in the thesis context. It needs to briefly clarify the purpose, the need, and the applicability of the solutions mentioned and also how these solutions have occurred.

A matrix summarizing the comparison among the selected case study buildings would be useful for supporting and illustrating further the conclusions.

The sentence starting in line 936 needs an extension/ clarification to be more specific.

Author Response

Firstly, thank you for taking the time to review my paper and provide feedback; it is much appreciated.

The initial draft was a reformatted transcript for the dissertation submitted for my masters degree. I have now comprehensively redrafted the paper in-line with the MDPI template. Many sections have been rewritten and superfluous paragraphs/figures have been removed, whilst retaining the core analysis and conclusions. Despite the changes, I have tried to address all your comments to the best of my ability below.

I have briefly expanded upon the case study selection. Principally, they are chosen for being similar heritage grades, completed at similar times (therefore addressing the same building standards), and having non-domestic functions. Factors of size and floor area are not particularly relevant in the scope of this article as the two case studies are not used for direct comparison. Instead, the methodology acknowledges successful sustainable design strategies from the superior case study (ECCI) to help speculate suitable proposals in the analysis of the other. 

The main analysis of the article in Chapters 3-4 is focussed on one case study (30 James Street) to analyse proposed sustainable design solutions. The results of this are then used in the discussion chapter to speculate areas of improvement for the existing standards. 

I have orthographic drawings of both case studies, however they were not included in article as they do not directly contribute to the discussion. Total floor area for the tested case study is given in Chapter 3 for the purpose of calculations.

Figure 17 is superfluous and has been removed.

A new sentence has been included to explain the additional solutions in chapter 6 (now chapter 4). Essentially, some sustainable design techniques are discovered to be financially unviable and solutions 3+4 revise solutions 1+2 to account for this.

A matrix comparing the two case studies is not needed. The ECCI study is only included in the article for the purpose of identifying alternative sustainable design measures not included in existing standards. However, figure 11 recaps measures identified from the ECCI and figure 17 summarises conclusions from the research in a proposed framework of revisions to the existing standards.

Line 936 has been rewritten in the new conclusion section. This paragraph ties directly into chapter 4.2 in the new draft.

Reviewer 2 Report

  1. Paper type. There is no such paper type as “Dissertation” for MDPI Sustainability Journal. Clearly the authors didn’t bother to do a simple search on article types. I wonder if the authors read any papers from Sustainability Journal at all?
  2. Detailed affiliation of both authors is missing. Why the hotmail email address, not official university email account has been provided? Who is the corresponding author on that work?
  3. Number of pages (51 pages) is problematic. Even it is mentioned on Sustainability Journal page that there is no restriction regarding the length of the paper, the authors should use a common sense and never propose such length of a paper. The only exception to the rule should be performing of systematic reviews by numerous authors. See following example: http://dx.doi.org/10.3390/su11226400I suggest to publish this work as a book, or shorten it to maximum 20-25 pages including the references and images and resubmit again for a proper review. Also, it would be advisable to use the  Microsoft Word template  for Sustainability Journal and exactly follow its format including referencing style. It seems to me that the authors just copied and pasted parts of the PhD work without reformatting it.

Author Response

Thank you for your feedback. I would like to start by explaining that I do not have a PhD, nor am I working towards one. I am a student of Architecture and this research was my dissertation, submitted as part of my masters degree. As such, I am not familiar with the templates and protocols of MDPI as I have never submitted a scientific article before.

I am aware that my initial submission was far from a final draft and this is largely because the article was not written with publication in mind. I have now put in significant effort to improve it and elevate it from a student's work into a legitimate article.

1. In fact I did bother to check article types, however I was unsure how this paper would classify as it was my student dissertation and did not strictly adhere to any one category. After restructuring the research I have listed it as an 'article', although if you have a suggestion for a more fitting title I will be happy to reconsider.

2. As I mentioned, this work constituted part of my masters degree. As I am no longer a student I am unable to use my university email. I could use my current work email, although I am likely to move on in the coming years, so I thought it best to include my personal email. Corresponding authors have also been included.

3. The length of the article has been reduced to 22 pages plus appendices after much time and effort spent rewriting and restructuring. Superfluous passages and figures have been removed and references reconsidered. Text and paragraphs now align exactly with the MDPI template.

Round 2

Reviewer 2 Report

Dear Authors,

I would like to congratulate you on great improvement of your manuscript. 

I have 3 minor comment.

  1. Email address. I suggest to use your current email address from Pollard Thomas Edwards Architects.
  2. Please add the copyright information for the images e.g.: “Source: authors’ own work” or “© by Andre Benz/Unsplash”
  3. Can you please add some more research papers to your reference list?.

Author Response

Thank you for your feedback.

  1. I have updated to my work email
  2. copyright included in figure/table titles
  3. I have managed to work in 5 more research papers throughout the article however I will have to leave it there to get it submitted for tomorrow as requested. Hopefully this is enough.